# Generalization in multitask deep neural classifiers: a statistical physics approach

**Tyler Lee**
Intel AI Lab
tyler.p.lee@intel.com

**Anthony Ndirango**
Intel AI Lab
anthony.ndirango@intel.com

## Abstract

A proper understanding of the striking generalization abilities of deep neural networks presents an enduring puzzle. Recently, there has been a growing body of numerically-grounded theoretical work that has contributed important insights to the theory of learning in deep neural nets. There has also been a recent interest in extending these analyses to understanding how multitask learning can further improve the generalization capacity of deep neural nets. These studies deal almost exclusively with regression tasks which are amenable to existing analytical techniques. We develop an analytic theory of the nonlinear dynamics of generalization of deep neural networks trained to solve classification tasks using softmax outputs and cross-entropy loss, addressing both single task and multitask settings. We do so by adapting techniques from the statistical physics of disordered systems, accounting for both finite size datasets and correlated outputs induced by the training dynamics. We discuss the validity of our theoretical results in comparison to a comprehensive suite of numerical experiments. Our analysis provides theoretical support for the intuition that the performance of multitask learning is determined by the noisiness of the tasks and how well their input features align with each other. Highly related, clean tasks benefit each other, whereas unrelated, clean tasks can be detrimental to individual task performance.

## 1 Introduction

Despite the remarkable string of successful results demonstrated by deep learning practitioners, we still do not have a clear understanding of how these models manage to generalize so well, effectively evading many of the intuitions expected from statistical learning theory. The enigma is further heightened when one considers multitask learning, especially in regimes where labeled data is scarce. In order to make specific assertions about the effective transfer of knowledge across tasks, one needs a predictive framework to address generalization in a multitask setting. There has been a noticeable uptick in recent efforts to build a rigorous theoretical foundation for deep learning (see, e.g. [1, 2, 3, 4, 5, 6, 7, 8, 9, 10] for a sampling of this trend). To the best of our knowledge (with one exception, described below), none of the existing analytical work deals with multitask learning.

Multitask learning holds promise for training more generalized and intelligent learning systems [11]. It comprises a broad set of strategies loosely defined by the presence of multiple objective functions and a set of shared parameters optimized for those objective functions. The most prevalent formulation of multitask learning in the literature is the addition of supervised auxiliary task(s) to assist in training a network to better perform a target task of interest (*main task*)[12, 13, 14, 15]. In this framework the only purpose of the auxiliary task(s) is to produce improved generalization performance on the main task. This benefit is thought to arise from an inductive bias placed on the learning of the main task towards learning more general features [11]. Since the features learned through multitask learning blend the optimal features for all of the optimized tasks, there is an

assumed dependence of the multitask benefit on the relatedness of the auxiliary tasks to the main task (e.g. if the optimal features for the auxiliary task are orthogonal to those of the main task, then the main task will be best optimized by ignoring the auxiliary task entirely). How exactly to define "relatedness" in the context of multitask learning in deep neural networks remains unknown. The most explicit definition to date, to our knowledge, comes from [16], where it is described as the angles between the singular vectors of the implicit input-output function learned by the network. While this definition is narrow, it lends a nice starting point for a theoretical analysis in the multitask setting. Outside of the work done in [16] on multitask learning in linear regression networks, the theory of multitask learning in neural networks remains unexplored. In this work we hope to further the theoretical understanding of multitask benefits to multiclass classification problems, a much more common class of problems in modern machine learning.

To narrow the scope of this study, we have chosen to focus on the formulation of multitask learning where the neural network is defined as having a single shared trunk and multiple task-specific heads. Many recent studies have sought to explore alternative methods of parameter sharing, though these do not usually lend themselves as easily to this form of theoretical analysis [17, 18]. Further, multitask learning also provides an interesting strategy for learning a single universal representation for many tasks possibly across multiple domains [19, 20, 21]. In this strategy there is often no clear "main" task and it is not clear that the benefit to be gained is even improved generalization performance on any of the trained tasks. Instead the benefit could be seen as improved performance over a set of problems given a fixed parameter budget or improved transfer learning to unseen tasks [22]. While these are certainly exciting research directions and could benefit from careful theoretical scrutiny, we leave them for future work.

This manuscript is structured as follows: in section 2 we describe the theory behind single task learning in classification networks. In section 3 we describe, both analytically and empirically, the training dynamics of such networks. In section 4 we extend this work to account for multitask learning of simple classification tasks. Finally, in section 5 discuss interesting leads and future directions.

## 2 Theoretical Underpinnings

A convenient framework for analyzing multitask problems was introduced in [16], addressing regression problems in deep linear neural networks. Given the success of that approach, could the techniques in [16] be generalized to deep neural net classifiers with softmax outputs? Our analysis provides an affirmative answer to this question, albeit at considerable technical cost: despite a strong conceptual similarity between analyzing regression and softmax classification problems, the structure of the solutions to the classification problem differ markedly from those obtained in the regression case. On the other hand, and perhaps unsurprisingly, the intuition gleaned from [16] about the conditions required for effective multitask learning carry over to the classification problems, in spite of the technical differences between the analysis of classification and regression tasks.

We adopt the student-teacher setup popularized several decades ago in early attempts to theoretically understand the generalization abilities of neural networks (see, e.g. [23]) and recently revisited in [16]. We will attempt to closely follow the notational conventions in [16] with the hope of establishing a common language for analyzing these sorts of problems. The key insight behind the analysis of softmax classifiers is the uncanny resemblance of the training dynamics of deep neural nets to the physical dynamics of disordered systems. In particular, we take advantage of a formal similarity between deep neural softmax classifers and a generalized version of Derrida's Random Energy Model (REM) [24]. A generalization of the REM is required because the outputs of a deep neural network are correlated random variables, in contrast to the i.i.d conditions that render the original REM solvable. Furthermore, deep learning practitioners do not work with infinite size models, so we also have to take into account finite size effects.

### 2.1 Teacher Network

Following [16], we consider low rank teacher networks which serve to provide a training signal to arbitrary student networks. We begin with a 3-layer teacher network defined by $\overline{N}_\ell$ units in layer $\ell$ and weight matrices $\overline{\mathbf{W}}^{21} \in \mathbb{R}^{\overline{N}_2 \times \overline{N}_1}$ between the input and hidden layer and $\overline{\mathbf{W}}^{32} \in \mathbb{R}^{\overline{N}_3 \times \overline{N}_2}$

between the hidden layer and an *argmax* output layer. We also define $\overline{\mathbf{W}} \equiv \overline{\mathbf{W}}^{32}\overline{\mathbf{W}}^{21} \in \mathbb{R}^{\overline{N}_3 \times \overline{N}_1}$ for the teacher's composite weight.

We consider teachers that produce noisy outputs using a noise perturbed composite weight matrix $\hat{\boldsymbol{\Sigma}} \equiv \overline{\mathbf{W}} + \boldsymbol{\xi}$, where $\boldsymbol{\xi} \in \mathbb{R}^{\overline{N}_3 \times \overline{N}_1}$ has i.i.d elements.

During training, the teacher network takes in an input data matrix $\mathbf{X} \in \mathbb{R}^{\overline{N}_1 \times N_{\text{data}}}$, and produces noisy vector outputs $\hat{\boldsymbol{y}} \equiv \underset{\text{over rows}}{\operatorname{argmax}} \{\hat{\boldsymbol{\Sigma}}\mathbf{X}\} \in \mathbb{R}^{N_{\text{data}}}$

thereby furnishing a rule for producing (noisy) labels $\hat{\boldsymbol{y}}$ from inputs $\mathbf{X}$. At test time, the student is tested against noise-free labels generated via $\overline{\mathbf{y}} \equiv \underset{\text{over rows}}{\operatorname{argmax}} \{\overline{\mathbf{W}}\mathbf{X}\} \in \mathbb{R}^{N_{\text{data}}}$

At this point, we take a slight departure from the setup in [16]: in their setup, the data matrix is taken to be orthonormal, whereas we take $\mathbf{X}$ to have entries drawn independently from a standard Gaussian distribution. Similarly, the elements of the noise matrix $\boldsymbol{\xi}$ are i.i.d centered normal variables with variance $\hat{\sigma}^2/\overline{N}_1$. The scale of $\hat{\sigma}$ is chosen in such a way that there is a non-zero probability for label-flipping, i.e. $\operatorname{Prob}(\hat{\boldsymbol{y}} \neq \overline{\mathbf{y}}) > 0$.

## 2.2 Student Network

We first consider a 3-layer student network. In general, the student network has the same number of input and output units as the teacher since these are defined by the specifics of the task at hand. However, the student has no knowledge of the teacher's internal architecture. Thus, the number of hidden units in the student's network will almost surely be different from the teacher's. Writing $N_2$ for the student's number of hidden units, we have student weight matrices $\mathbf{W}^{21} \in \mathbb{R}^{N_2 \times \overline{N}_1}$ between the input and hidden layer and $\mathbf{W}^{32} \in \mathbb{R}^{\overline{N}_3 \times N_2}$ between the hidden layer and the softmax output layer. We also define $\mathbf{W} \equiv \mathbf{W}^{32}\mathbf{W}^{21} \in \mathbb{R}^{\overline{N}_3 \times \overline{N}_1}$ for the student's composite weight.

Given an input data matrix $\mathbf{X} \in \mathbb{R}^{\overline{N}_1 \times N_{\text{data}}}$, the student computes a matrix output

$$\mathbf{Y}(\mathbf{W}\mathbf{X}) = \operatorname{softmax}(\mathbf{W}\mathbf{X})$$

Note that $\mathbf{Y} \in \mathbb{R}^{\overline{N}_3 \times \overline{N}_1}$ is a matrix with elements

$$\mathbf{Y}_{c\mu}(\mathbf{W}\mathbf{X}) = \operatorname{softmax}\left(\sum_{k=1}^{\overline{N}_1} \mathbf{W}_{ck}\mathbf{X}_{k\mu}\right), \qquad 1 \leq c \leq \overline{N}_3,\, 1 \leq \mu \leq N_{\text{data}}$$

which is interpreted as the probability that the student assigns a class label $c$ given an input $\mathbf{x}^\mu$ drawn from the $\mu^{\text{th}}$ column of $\mathbf{X}$.

The student is trained by minimizing a cross-entropy loss

$$\mathcal{L}_{\text{train}} = -\frac{1}{N_{\text{data}}} \sum_{\mu=1}^{N_{\text{data}}} \sum_{c=1}^{\overline{N}_3} \delta_{c,\hat{\boldsymbol{y}}_\mu(\mathbf{X})} \ln \mathbf{Y}_{c\mu}(\mathbf{W}\mathbf{X}), \qquad \text{(where $\delta$ is the Kronecker delta.)} \quad (1)$$

## 3 Training Dynamics: Theory v/s Experiment

We use vanilla SGD to train the student network. A detailed derivation of the dynamics of training is presented in appendix A. The relevant equations are given by

$$\begin{aligned} \tau\frac{d}{dt}\mathbf{W}^{32} &= \left(\mathbf{G}(\hat{\boldsymbol{\Sigma}})\hat{\boldsymbol{\Sigma}} - \mathbf{G}(\mathbf{W})\mathbf{W}\right)\mathbf{W}^{21^T} \\ \tau\frac{d}{dt}\mathbf{W}^{21} &= \mathbf{W}^{32^T}\left(\mathbf{G}(\hat{\boldsymbol{\Sigma}})\hat{\boldsymbol{\Sigma}} - \mathbf{G}(\mathbf{W})\mathbf{W}\right) \end{aligned} \quad (2)$$

where $1/\tau$ is the SGD learning rate, and $\mathbf{G} : \mathbb{R}^{\overline{N}_3 \times \overline{N}_1} \mapsto \mathbb{R}^{\overline{N}_3 \times \overline{N}_3}$ is a non-linear, *positive semi-definite* matrix-valued function which captures the gradient of the softmax function averaged over the training data (see appendix A:13 for a precise definition). The solutions to (2) are very different from those obtained for the regression case in [16].

Further insight into the dynamics (2) is provided by considering the so-called *training aligned* (TA) case as defined in [16] where one initializes the student's weights such that the initial value of the student's composite weight is $\mathbf{W}_0 = \hat{\boldsymbol{U}} \mathbf{S}_0 \hat{\boldsymbol{V}}^T$ given the noisy teacher's SVD $\hat{\boldsymbol{\Sigma}} = \hat{\boldsymbol{U}} \, \hat{\boldsymbol{S}} \, \hat{\boldsymbol{V}}^T$, where $\mathbf{S}_0$ is the student's initial singular value matrix.

A detailed analysis of the TA dynamics is presented in full generality in appendix B. For a rank one teacher in the TA case, i.e. if the noisy teacher's SVD is $\hat{\boldsymbol{\Sigma}} = \hat{s} \hat{\boldsymbol{u}} \hat{\boldsymbol{v}}^T$, equation (2) simplifies further to an equation for the student's largest singular value, with all the other singular values exponentially suppressed in time. Explicitly, writing $s \equiv \max \mathbf{S}$ for the student's largest singular value, equation (2) becomes

$$\tau \frac{d}{dt} s = 2s\hat{\boldsymbol{u}} \cdot \left( \hat{s} \mathbf{G}(\hat{s}\hat{\boldsymbol{u}}\hat{\boldsymbol{v}}^T) - s\mathbf{G}(s\hat{\boldsymbol{u}}\hat{\boldsymbol{v}}^T) \right) \hat{\boldsymbol{u}} \tag{3}$$

Numerically integrating equation (3) yields the graphs shown in Figure 1. The figure reveals excellent agreement between theory and experiment over a wide range of initial conditions.

## 4 Multitask Generalization Dynamics: Theory v/s Experiment

### 4.1 Teacher Networks

In the multitask setting, we have two teacher networks represented by $\overline{N}_3 \times \overline{N}_1$ weight matrices $\overline{\mathbf{W}}_A$ and $\overline{\mathbf{W}}_B$ with ranks $\overline{N}_2^A$ and $\overline{N}_2^B$ respectively. Their noise-perturbed versions, $\hat{\boldsymbol{\Sigma}}_A$, $\hat{\boldsymbol{\Sigma}}_B$ are defined as before, so that the teachers produce noisy labels $\hat{\boldsymbol{y}}_{A/B} \equiv \underset{\text{over rows}}{\mathrm{argmax}} \left\{ \hat{\boldsymbol{\Sigma}}_{A/B} \mathbf{X} \right\}$ and noise free labels $\overline{\mathbf{y}}_{A/B} \equiv \underset{\text{over rows}}{\mathrm{argmax}} \left\{ \overline{\mathbf{W}}_{A/B} \mathbf{X} \right\}$.

### 4.2 Student Network

In the multitask setting, a composite student network is designed to learn multiple tasks jointly from the teachers. In general, the student network will consist of a trunk comprised of a stack of hidden layers shared across tasks, augmented by a set of specialized heads specific to individual tasks. This setup is identical to the one used in [16].

For three-layer students, we continue to denote the trunk's composite weight matrix by $\mathbf{W}^{21}$ and write $\mathbf{W}_A^{32}$, $\mathbf{W}_B^{32}$ for the weights in the heads, and $\mathbf{W}_A \equiv \mathbf{W}_A^{32}\mathbf{W}^{21}$, $\mathbf{W}_B \equiv \mathbf{W}_B^{32}\mathbf{W}^{21}$ for the corresponding composite weights. Note that, crucially, both students share the trunk weights $\mathbf{W}^{21}$.

The students are trained to minimize a weighted sum of the cross-entropy losses pertaining to each task, i.e. $\mathcal{L} = \alpha_A \mathcal{L}_A + \alpha_B \mathcal{L}_B$. In general, the weighting coefficients $\alpha_A$, $\alpha_B$ can be chosen via some optimization method or even learned as part of the model's training procedure. However, we will only consider the simplest case where $\alpha_A = \alpha_B = 1$.

We arbitrarily pick task A as the main task that we're interested in, and consider task B as an auxiliary task whose sole purpose is to improve the performance of task A. We are thus interested in finding out what properties of task B are required in order to improve the student's learning of task A. This naturally leads to the idea of *task-relatedness*, a well-known, though loosely-defined, concept in the literature on multitask learning [11].

### 4.3 Task Relatedness

As noted in the introduction, we currently lack a precise definition of task-relatedness in the context of multitask learning in deep neural networks. The authors of [16] propose defining task-relatedness as a function of the angles between the singular vectors of the implicit input-output function learned by

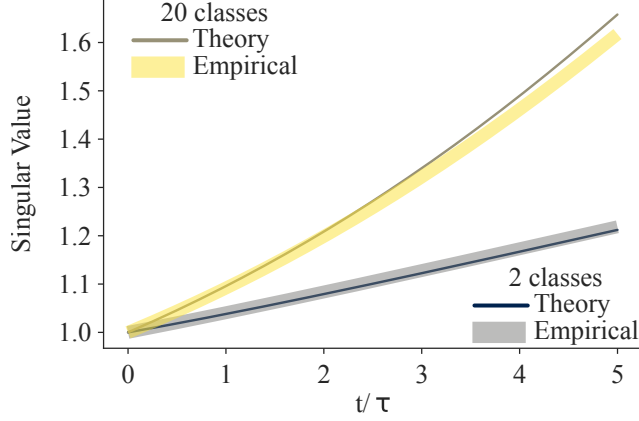

Figure 1: Comparing the theoretical predictions in (3) to empirical results. $1/\tau = 10^{-3}$ is the learning rate, so the figure shows training for 5k steps (chosen as the minimum of the validation error). The empirical results are obtained using 10 different random seeds. The results shown are for a 2-class and 20-class classification task using 100 training data points to highlight the fact that the theory agrees with experiment over a wide range of class sizes.

the network. As it turns out, as a direct consequence of the SGD dynamics in (2), the same definition appears naturally in the student-teacher framework for multitask classifiers.

Given two tasks $A$ and $B$ defined by two teachers with weight matrices $\overline{\mathbf{W}}_A$ and $\overline{\mathbf{W}}_B$ respectively, we denote their SVDs by $\overline{\mathbf{W}}_{A/B} = \overline{\mathbf{U}}_{A/B} \overline{\mathbf{S}}_{A/B} \overline{\mathbf{V}}_{A/B}^T$. We define the relatedness $\boldsymbol{r}_{AB}$ between tasks $A$ and $B$ as

$$\boldsymbol{r}_{AB} := \overline{\mathbf{V}}_B^T \overline{\mathbf{V}}_A \tag{4}$$

## 4.4 Multitask Benefit

Table 1: Key takeaways from multitask analysis

| | independent variables | | | effect on $MT_{A \leftarrow B}$ | analytical explanation |
|---|---|---|---|---|---|
| | $r_{AB}$ | $\bar{s}_B$ | $N_{\text{data}}$ | | |
| (a) | 0 | any | any | 0 | $s_A = \widetilde{s}_A$ |
| (b) | $> 0$ | $\nearrow$ | any | $\nearrow$ | $(s_A - \widetilde{s}_A) \searrow$ as $\bar{s}_B \nearrow$ |
| (c) | $r_{AB} \nearrow (0 < r_{AB} \ll 1)$ | any | limited | $\nearrow$ | appendix:C.1, eqn. (36) |
| (d) | any | any | abundant | small | $\widetilde{s}_A g(\widetilde{s}_A) \to \bar{s}_A g(\bar{s}_A)$ |

For the purposes of quantifying any gains in performance from multitask learning relative to models trained on a single task, we introduce the notion of a *multitask benefit*. We arrive at our multitask benefit by comparing the optimal performance of the multitask model on the main task, say $A$ to the optimal performance of a *baseline* model trained only on task $A$.

Given the multitask generalization loss $\mathcal{L}_{AB} = \mathcal{L}_A + \mathcal{L}_B$, we define $\mathcal{L}_{A|B} := \mathcal{L}_{AB} - \mathcal{L}_B$ as the generalization loss on task $A$ when task $A$ is trained jointly with task $B$. This quantity is to be compared to the generalization loss $\widetilde{\mathcal{L}}_A$ defined as the loss when task A is trained on its own. Following [16], we define the multitask benefit conferred on task A by task B via

$$MT_{A \leftarrow B} \equiv \min_t \left\{ \widetilde{\mathcal{L}}_A(t) \right\} - \min_t \left\{ \mathcal{L}_{A|B}(t) \right\}$$

Remarkably, one can place a tight bound on the multitask benefit using a relatively simple argument based on the concavity of the logarithm function. We present here the result for the simpler case of a TA model with rank one teachers and relegate the general case to appendix C. For a TA model with rank one teachers with SVD $\overline{\mathbf{W}}_A = \bar{s} \boldsymbol{u}_A \boldsymbol{v}_A^T$, we abbreviate $g(s) := \boldsymbol{u}_A \cdot \mathbf{G}(s \boldsymbol{u}_A \boldsymbol{v}_A^T) \boldsymbol{u}_A \geq 0$,

with $\mathbf{G}$ as featured in the training dynamics in equation (2) and defined in appendix A:13. The key takeaways of this analysis are summarized in Table 1 and described more fully below.

As derived in Appendix C (cf. equations C:24 and C:25), the bound on the multitask benefit is

$$\left(s_A - \widetilde{s}_A\right)\left(\overline{s}_A g(\overline{s}_A) - s_A g(s_A)\right) \leq MT_{A \leftarrow B} \leq \left(s_A - \widetilde{s}_A\right)\left(\overline{s}_A g(\overline{s}_A) - \widetilde{s}_A g(\widetilde{s}_A)\right) \quad (5)$$

Notice that the factor $\left(\overline{s}_A g(\overline{s}_A) - \widetilde{s}_A g(\widetilde{s}_A)\right)$ on the RHS of equation (5) depends only quantities pertaining to the baseline single task case, and hence is entirely independent of the training dynamics of the multitask case.

In contrast, the sign of $(s_A - \widetilde{s}_A)$ depends on the multitask teachers' singular values for tasks A and B, their corresspponding SNRs, and the relatedness $r_{AB}$ between tasks A and B (see the discussion surrounding equations 28-37 in Appendix C.1). For unrelated tasks, *viz.* $r_{AB} = 0$, one obtains $s_A = \widetilde{s}_A$ (cf. C.1:28) and so the multitask benefit vanishes. For "weakly related" tasks, *viz.* $0 < r_{AB} \ll 1$, (C.1:35) shows that high SNR auxiliary tasks have a deleterious effect on $MT_{A \leftarrow B}$.

In the high SNR regime, the noisy teacher's singular values are larger than the noise-free case. Since the student's dynamics is driven by the noisy teacher, $s_A \to \hat{s}_A \geq \overline{s}_A$ in the high SNR regime. Under these conditions, equation (C.1:31) implies that $MT_{A \leftarrow B} \geq 0$.

In the low SNR regime, the noisy teacher's singular values lie in the bulk of the MP sea [25]. In this case, the student's dynamics is driven by noise, so that $s_A \to \hat{s}_A < \overline{s}_A$ for low SNRs. Under these conditions, a positive $MT_{A \leftarrow B}$ occurs only if the constraints on $r_{AB}$ and $\overline{s}_B$ leading to equation (C.1:33) are satisfied.

In regimes where labeled training data is abundant, the factor $\left(\overline{s}_A g(\overline{s}_A) - \widetilde{s}_A g(\widetilde{s}_A)\right) \to 0$ in which case $MT_{A \leftarrow B} \to 0$, regardless of the relatedness between tasks (cf. equation C.1:37).

To summarize, the TA model predicts that multitask learning will have the largest impact under conditions mimicking scarce labeled data such that the baseline model underperforms on the main task, as long as the auxiliary tasks have some relatedness to the main task. Thus, coming up with auxiliary tasks that have a high degree of relatedness to the main task will be crucial to observing a positive multitask benefit.

While the results in this section have only been demonstrated for the special case of TA models, we will shortly see that the predictions are realized empirically in a wide variety of scenarios.

## 4.5  Data vs model uncertainty

Using the framework described above, we set out to describe the relationship between multitask benefit and several key factors that influence training of both the single task baseline - the amount and quality of the main task data - and multitask training - the amount, quality and relatedness of auxiliary task data. We systematically varied[1] these factors and computed the multitask benefit for 5 different training datasets, the results of which are summarized in Figure 2. To ensure that we had roughly class-balanced training datasets, we fixed $\overline{N}_3 = \overline{N}_2$, and set both to 10 for the experiments here. Other values for the rank showed similar results and data for rank 3 teacher networks can be found in Figure A2. The signal-to-noise ratio (SNR) of the data in each dataset is directly proportional to the singular value of the teacher network that generated each task's data.

We kept all singular values for a given teacher network the same and varied this value from .01 to 100. Similarly, we fixed the relatedness of teacher network B to $\overline{\mathbf{V}}_B^T \overline{\mathbf{V}}_A = r_{AB} I$, such that the singular vectors $\overline{V}_B$ were orthogonal to $\overline{V}_A$ with constant inner product. We varied this value from 0 to 1. This work demonstrates several interesting dependencies:

1. Multitask benefit increases with increasing task relatedness and SNR of the auxiliary data. This mirrors the finding from row (b) of Table 1.

2. Unrelated, high SNR auxiliary tasks are actually destructive to the learning process of the main task. Our theoretical framework provides an explanation for this observation in C.1:35.

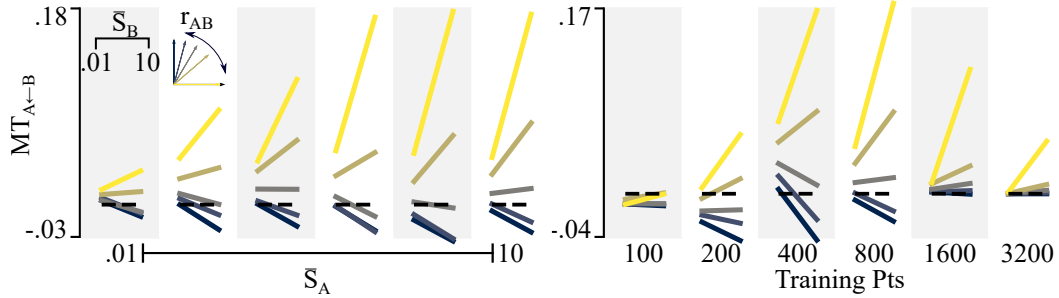

Figure 2: (Left) Summary of multitask benefits gained when the student network was trained with increasing signal-to-noise ratio (SNR). With constant noise levels, the SNR increases with the singular values for teacher A, $\overline{S}_A$, were increased from .01 to 10 (alternating stripes, left-to-right). For each value of $\overline{S}_A$ (x-axis), the average multitask benefit was computed for low SNR auxiliary tasks ($\overline{S}_B$) and high SNR auxiliary tasks (each line segment, left-to-right) across 5 levels of task relatedness ($r_{AB}$). Data is plotted for 800 training points. This demonstrates that multitask benefit is correlated with task relatedness and SNR for related tasks, yet negatively correlated with SNR for unrelated tasks. (Right) Summary of multitask benefits with increasing amount of training data (alternating stripes, left-to-right). At 100 training points the network still struggles to train and does not gain a generalization benefit from auxiliary data. For > 200 training points, the network begins to leverage the related auxiliary data to improve performance. When the dataset is very large, performance nearly reaches its ceiling and the auxiliary data has little effect. See Figure A1 for the complete set of interactions among these variables.

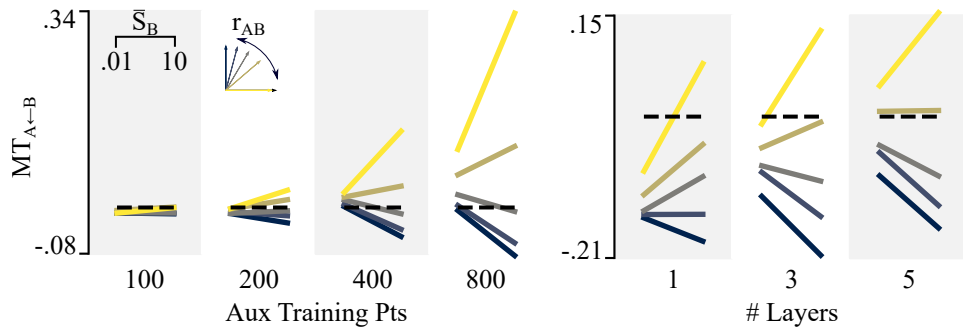

Figure 3: (Left) Summary of multitask benefits gained when the student network was trained with increasing amounts of auxiliary task data . For each quantity of auxiliary task data (x-axis), the average multitask benefit was computed for low SNR aux tasks and high SNR aux tasks (each line segment, left-to-right) across 5 levels of task relatedness. All the data shown is for high SNR main tasks, and demonstrates that increasing relatedness and auxiliary task data give large multitask benefits. For more details see Figure A3. (Right) Summary of multitask benefits gained for nonlinear student networks of increasing depth (x-axis). Deeper nonlinear networks show similar trends to shallow linear networks. For more details see Figure A4.

In contrast, unrelated, noisy auxiliary tasks are readily ignored. This mirrors the findings from rows (a) and (c) of Table 1.

3. The main task must have a certain level of base performance either from clean data or larger amounts of data before multitask learning can help. This holds up to the point where single task performance nears optimal performance on the main task, as is the case when the amount of training data supplied is large. These statements mirror the findings from rows (c) and (d) of Table 1.

### 4.6 Auxiliary task data efficiency

Multitask learning is a popular strategy for extending the utility of a limited amount of main task data. This is often an interesting choice when auxiliary task data is easy to come by but main task data is expensive. To gauge the value of additional auxiliary task data while holding main task data fixed, we trained multitask student networks on 100 main task data points and up to 800 auxiliary task data points. These results are summarized in Figure 3 (left) and full results can be found in Figure A3. As auxiliary task data quantities increase we see similar trade-offs to those above, where related, high quality data provides a large multitask benefit, while unrelated, high quality data proves increasingly detrimental.

### 4.7 Multitask learning in deeper, nonlinear student networks

To ensure that our results can generalize to nonlinear and deeper networks, we varied the number of hidden layers in the student network and included a ReLU nonlinearity between each hidden layer. While this situation does not lend itself to clean theoretical analysis, we found that these networks behave qualitatively similar to the linear network results described above. These results are summarized in Figure 3 (right) and full results can be found in Figure A4. Again, multitask benefit is strongly correlated with relatedness and the SNR of both datasets. Interestingly, there is a general shift downwards in multitask benefit, suggesting that nonlinear networks require more highly related tasks in order to generate a significant performance increase.

## 5 Discussion and future directions

Here we demonstrate that, for linear classifier networks with a softmax output nonlinearity, generalization performance can be computed analytically. We extend the analysis in [16] to classification problems and show both theoretically and empirically that improvements from multitask learning are heavily related to training set size, task relatedness, and the noise levels inherent in the data. Networks given sufficient data to train well show improved performance when supplemented with related, high signal-to-noise ratio auxiliary tasks. Unrelated auxiliary tasks show little benefit and can be actively detrimental if they provide a strong enough training signal.

The problem of increasing the range of parameters from which one gets a multitask benefit and decreasing potential harms has received increasing interest in recent years, often through clever loss or gradient weighting strategies [26, 27, 28]. A careful interrogation of (5) should provide some insight on methods for maximizing the possible multitask benefit, a direction we leave for future work. Additionally, we have shown that our results generalize to deeper, more nonlinear student networks, though these networks are still quite different from networks used in practice. We expect the insights gained in this work, especially with regard to the critical properties of main and auxiliary task datasets will generalize well to more complex networks. Generalizing our results regarding task relatedness poses an interesting challenge for future research.

#### Acknowledgments

We would like to thank Cory Stephenson, Gokce Keskin, Oguz Elibol, Suchismita Padhy, and Ting Gong for many fruitful discussions regarding this work. We must also acknowledge Nicholas Sapp for his work in establishing the compute infrastructure that made the empirical portions of this work possible.

## Footnotes

[1]Code supporting this paper is available upon request

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
