[Supplementary Material]

# Appendices

## Notation

- Given a matrix $\mathbf{A}$, we will denote its transpose by $\mathbf{A}^{\dagger}$.
- Given a pair of random vectors $\vec{X}, \vec{Y}$, we will denote their cross covariance matrix by $\mathbf{C}_{YX}$.
- Given a pair of vectors $\vec{u} \in \mathbb{R}^m$, $\vec{v} \in \mathbb{R}^n$, we define $\vec{u} \otimes \vec{v} \in \mathbb{R}^{m \times n}$ as the matrix with entries $(\vec{u} \otimes \vec{v})_{ak} := u_a v_k$.
- Given a pair $\vec{u}, \vec{v}$ of $n$-dimensional vectors we denote their Hadamard product by $\vec{u} \odot \vec{v} \in \mathbb{R}^n$, i.e. $(\vec{u} \odot \vec{v})_k := u_k v_k$.
- $O(N) :=$ group of $N \times N$ orthogonal matrices.

## A  Teacher-Student Setup

### A.1  Teacher Network

We consider teachers defined by a weight matrix $\overline{\mathbf{W}} \in \mathbb{R}^{N_{\text{classes}} \times N_f}$, where $N_{\text{classes}}$ is the number of classes and $N_f$ the number of input features. Noisy teachers are defined by a weight matrix $\hat{\mathbf{\Sigma}} \equiv \overline{\mathbf{W}} + \boldsymbol{\xi}$, where $\boldsymbol{\xi} \in \mathbb{R}^{N_{\text{classes}} \times N_f}$ has entries drawn independently from a centered Gaussian distribution with variance $\hat{\sigma}^2 / N_f$.

During training, the teacher network takes in an input data matrix $\mathbf{X} \in \mathbb{R}^{N_f \times N_{\text{data}}}$, and produces noisy vector outputs

$$\hat{\boldsymbol{y}} \equiv \underset{\text{over rows}}{\operatorname{argmax}} \left\{ \hat{\mathbf{\Sigma}} \mathbf{X} \right\} \in \mathbb{R}^{N_{\text{data}}}$$

thereby furnishing a rule for producing (noisy) labels $\hat{\boldsymbol{y}}$ from inputs $\mathbf{X}$. At test time, the student is tested against noise-free labels generated via $\overline{\mathbf{y}} \equiv \underset{\text{over rows}}{\operatorname{argmax}} \left\{ \overline{\mathbf{W}} \mathbf{X} \right\} \in \mathbb{R}^{N_{\text{data}}}$.

The columns of $\mathbf{X}$ form a collection of $N_{\text{data}}$ feature vectors $\{\vec{X}^\mu\}$, $\mu = 1, \cdots, N_{\text{data}}$, drawn from a centered Gaussian distribution with covariance $\mathbf{C}_X$. We will write $\hat{y}(\vec{X}^\mu)$ for the label assigned to the feature vector $\vec{X}^\mu$. We assume that the matrix $\mathbf{X}$ is of full rank so that $\mathbf{X}^\dagger \mathbf{X}$ is invertible.

### A.2  Student Network

A student network with $L$ layers is defined via a collection of weight matrices $\mathbf{W}^{(l)} \in \mathbb{R}^{N_l \times N_{l-1}}$, $1 \leq l \leq L$, with $N_0 = N_f$ and $N_L = N_{\text{classes}}$. The student's composite weight matrix is given by $\mathbf{W} := \mathbf{W}^{(L)} \mathbf{W}^{(L-1)} \cdots \mathbf{W}^{(1)}$.

Define

$$
\begin{aligned}
\mathbf{W}_{>}^{(l)} &:= \mathbf{W}^{(L)} \cdots \mathbf{W}^{(l+1)}, \\
\mathbf{W}_{<}^{(l)} &:= \mathbf{W}^{(l-1)} \cdots \mathbf{W}^{(1)}
\end{aligned}
$$

so that, for $2 \leq l < L$,

$$\mathbf{W} = \mathbf{W}_{>}^{(l)} \mathbf{W}^{(l)} \mathbf{W}_{<}^{(l)}.$$

In particular, the gradient of any scalar valued function $f(\mathbf{W})$, with respect to $\mathbf{W}^{(l)}$ is given by

$$\nabla_{\mathbf{W}^{(l)}} f = \mathbf{W}_{>}^{(l)^{\dagger}} (\nabla_{\mathbf{W}} f) \mathbf{W}_{<}^{(l)^{\dagger}} \tag{6}$$

Let $\mathbb{P}_c(\mathbf{W}\vec{X})$ define the probability of observing class $c$ given $\mathbf{W}\vec{X}$. For a neural classifier, this reads

$$\mathbb{P}_c(\mathbf{W}\vec{X}) := \text{Probability}\left(\text{class } c \,\middle|\, \mathbf{W}\vec{X}\right) := \text{softmax}[\mathbf{W}\vec{X}][c].$$

The cross-entropy loss between the teacher's one-hot-distributed labels $\{\hat{y}(\vec{X}^\mu))\}$ and the student's softmax outputs can be written as

$$
\begin{aligned}
\mathcal{L}_{\text{train}}(\mathbf{W}|\hat{\boldsymbol{\Sigma}}, \mathbf{X}) &= -\frac{1}{N_{\text{data}}} \sum_{\mu=1}^{N_{\text{data}}} \ln \mathbb{P}_{\hat{y}(\vec{X}^\mu)}(\mathbf{W}\vec{X}^\mu) \\
&= -\frac{1}{N_{\text{data}}} \sum_{\mu=1}^{N_{\text{data}}} \sum_{c=1}^{N_{\text{classes}}} \mathbb{P}_c(\beta\hat{\boldsymbol{\Sigma}}\vec{X}^\mu) \ln \mathbb{P}_c(\mathbf{W}\vec{X}^\mu)
\end{aligned}
$$

where $\beta \gg 1$ is a parameter chosen such that $\mathbb{P}_c(\beta\hat{\boldsymbol{\Sigma}}\vec{X}^\mu)$ is arbitrarily close to the noisy teacher's outputs $\hat{y}(\vec{X}^\mu)$.

### A.3 Training Dynamics

The student's weights are updated layerwise via SGD. Adopting the "continuous time" version of SGD for ease of exposition, and using the identity (6), the layerwise update equations read

$$
\begin{aligned}
\tau\frac{d}{dt}\mathbf{W}^{(l)} &= -\nabla_{\mathbf{W}^{(l)}}\mathcal{L}_{\text{train}}(\mathbf{W}|\hat{\boldsymbol{\Sigma}}, \mathbf{X}) \\
&= -\mathbf{W}_>^{(l)\dagger} \left[ \frac{1}{N_{\text{data}}} \sum_{\mu=1}^{N_{\text{data}}} \sum_{c=1}^{N_{\text{classes}}} \mathbb{P}_c(\beta\hat{\boldsymbol{\Sigma}}\vec{X}^\mu) \left( \nabla_{\mathbf{W}} \ln \mathbb{P}_c(\mathbf{W}\vec{X}^\mu) \right) \right] \mathbf{W}_<^{(l)\dagger}
\end{aligned}
$$

A straightforward calculation reveals that

$$
\frac{1}{N_{\text{data}}} \sum_{\mu=1}^{N_{\text{data}}} \sum_{c=1}^{N_{\text{classes}}} \mathbb{P}_c(\beta\hat{\boldsymbol{\Sigma}}\vec{X}^\mu) \left( \nabla_{\mathbf{W}} \ln \mathbb{P}_c(\mathbf{W}\vec{X}^\mu) \right) = \mathbf{C}_{YX}(\mathbf{W}) - \mathbf{C}_{YX}(\beta\hat{\boldsymbol{\Sigma}})
$$

where the matrix $\mathbf{C}_{YX}(\mathbf{W})$ defined by

$$
\mathbf{C}_{YX}(\mathbf{W})_{c,k} = \frac{1}{N_{\text{data}}} \sum_{\mu=1}^{N_{\text{data}}} \mathbb{P}_c(\mathbf{W}\vec{X}^\mu)X_k^\mu \tag{7}
$$

is the student's estimate of the empirical cross-covariance between the softmax outputs and the feature vectors determined using the training dataset. Similarly, $\mathbf{C}_{YX}(\beta\hat{\boldsymbol{\Sigma}})$ is the empirical cross-covariance between the feature vectors and the labels generated by the teacher.

Therefore,

$$
\tau\frac{d}{dt}\mathbf{W}^{(l)} = \mathbf{W}_>^{(l)\dagger} \left[ \mathbf{C}_{YX}(\beta\hat{\boldsymbol{\Sigma}}) - \mathbf{C}_{YX}(\mathbf{W}) \right] \mathbf{W}_<^{(l)\dagger} \tag{8}
$$

Equation (8) yields an interesting relationship between the weights in consecutive layers, *viz.*

$$
\frac{d}{dt}\left[ \mathbf{W}^{(l+1)\dagger}\mathbf{W}^{(l+1)} \right] = \frac{d}{dt}\left[ \mathbf{W}^{(l)}\mathbf{W}^{(l)\dagger} \right] \qquad 1 \le l \le L-1 \tag{9}
$$

Using equation (8), a straightforward calculation gives

$$
\tau\frac{d}{dt}\mathcal{L}_{\text{train}}(\mathbf{W}|\hat{\boldsymbol{\Sigma}}, \mathbf{X}) = -\sum_{l=1}^{L} \text{Tr}\left( \mathbf{W}_<^{(l)\dagger}\mathbf{W}_<^{(l)} \left[ \mathbf{C}_{YX}(\beta\hat{\boldsymbol{\Sigma}}) - \mathbf{C}_{YX}(\mathbf{W}) \right]^\dagger \mathbf{W}_>^{(l)}\mathbf{W}_>^{(l)\dagger} \left[ \mathbf{C}_{YX}(\beta\hat{\boldsymbol{\Sigma}}) - \mathbf{C}_{YX}(\mathbf{W}) \right] \right) \tag{10}
$$

Each summand on the RHS of equation (10) is the trace of a product of symmetric positive semi-definite matrices. Hence $\frac{d\mathcal{L}_{\text{train}}}{dt} \leq 0$ throughout training. Thus, SGD is guaranteed to converge to a solution which minimizes $\mathcal{L}_{\text{train}}(\cdot|\hat{\boldsymbol{\Sigma}}, \mathbf{X})$, although we have not provided any information about the rate of convergence.

Furthermore,

$$\min_{\mathbf{W}} \mathcal{L}_{\text{train}}(\mathbf{W}|\hat{\boldsymbol{\Sigma}}, \mathbf{X}) = \mathcal{L}_{\text{train}}(\hat{\boldsymbol{\Sigma}}|\hat{\boldsymbol{\Sigma}}, \mathbf{X}).$$

and

$$\mathbf{W} \text{ is a minimum of } \mathcal{L}_{\text{train}}(\cdot|\hat{\boldsymbol{\Sigma}}, \mathbf{X}) \;\Leftrightarrow\; \mathbf{C}_{YX}(\mathbf{W}) = \mathbf{C}_{YX}(\beta\hat{\boldsymbol{\Sigma}}).$$

In other words, the optimal solutions include all cases where the student's estimate of the empirical cross-covariance matches that of the noisy teacher. The number of such solutions is highly degenerate due to the fact that the softmax function is invariant under all transformations $\mathbf{W} \to \mathbf{W} + \vec{1} \otimes \vec{v}$ for any vector $\vec{v} \in \mathbb{R}^{N_f}$, where $\vec{1} \in \mathbb{R}^{N_{\text{classes}}}$ is the vector of all ones.

A straightforward computation shows that the Hessian of $\mathcal{L}_{\text{train}}(\cdot|\hat{\boldsymbol{\Sigma}}, \mathbf{X})$ has only non-negative eigenvalues, which combined with equation (10) leads to the conclusion that the set of minima of the loss is given by

$$\left\{ \mathbf{W} = \beta\hat{\boldsymbol{\Sigma}} + \lambda \sum_{\mu=1}^{N_{\text{data}}} \vec{1} \otimes (\mathbf{X}^{\dagger}\mathbf{X})^{-1}\vec{X}^{\mu} \;\; \forall \lambda \in \mathbb{R} \right\}. \tag{11}$$

### A.3.1 Training in the limit of infinite data

Finally, we note that as $N_{\text{data}} \to \infty$, equation (7) reads

$$\lim_{N_{\text{data}} \to \infty} \frac{1}{N_{\text{data}}} \sum_{\mu=1}^{N_{\text{data}}} \mathbb{P}_c(\mathbf{W}\vec{X}^{\mu})X_k^{\mu} \to \mathbb{E}\left( X_k \mathbb{P}_c(\mathbf{W}\vec{X}) \right) \tag{12}$$

where $\mathbb{E}(\cdot)$ denotes the expectation over $\vec{X}$. When $\vec{X}$ is a centered Gaussian random vector with covariance $\mathbf{C}_X$, then Gaussian integration by parts in (12) yields

$$\mathbb{E}\left( X_k \mathbb{P}_c(\mathbf{W}\vec{X}) \right) = [\mathbf{G}(\mathbf{W})\mathbf{W}\mathbf{C}_X]_{ck}$$

where the matrix $\mathbf{G}(\mathbf{W})$ is defined as

$$\mathbf{G}(\mathbf{W})_{c,c'} := \mathbb{E}\left( \mathbb{P}_c(\mathbf{W}\vec{X}) \right) \delta_{c,c'} - \mathbb{E}\left( \mathbb{P}_c(\mathbf{W}\vec{X})\mathbb{P}_{c'}(\mathbf{W}\vec{X}) \right). \tag{13}$$

Thus, from equations (7) and (12), we obtain

$$\lim_{N_{\text{data}} \to \infty} \mathbf{C}_{YX}(\mathbf{W}) = \mathbf{G}(\mathbf{W})\mathbf{W}\mathbf{C}_X. \tag{14}$$

We note that, from the definition in (13), $\mathbf{G}(\mathbf{W})$ is a positive semi-definite matrix with a single zero eigenvalue. Furthermore, if the diagonal entries of $\mathbf{W}\mathbf{W}^{\dagger}$ are much larger in magnitude than its off diagonal entries, then one can combine the HLP theorem [29] with a generalization of Derrida's REM techniques [24] to show that:

1.
$$\mathbf{G}(\mathbf{W}) \simeq \frac{1}{N_{\text{classes}} - 1} \text{Tr}(\mathbf{G}(\mathbf{W})) \left( \mathbf{I} - \frac{1}{N_{\text{classes}}}\vec{1} \otimes \vec{1} \right).$$

2. If the SVD of $\mathbf{W}$ is given by $\mathbf{W} = \mathbf{USV}^\dagger$, then $\mathrm{Tr}(\mathbf{G}(\mathbf{W})) \simeq g(\mathbf{S})$ where the explicit functional form of the real-valued function $g$ can be accurately estimated for large values of the norm $\|\mathbf{S}\|$ of $\mathbf{S}$. Under the stated conditions, one can show that

$$\text{the individual components, } g(\boldsymbol{S})S_{\alpha\alpha}, \text{ decrease monotonically with the norm } \|\boldsymbol{S}\|. \quad (15)$$

3. Furthermore,

$$\mathbf{U}^\dagger \mathbf{G}(\mathbf{USV}^\dagger)\mathbf{U} \simeq \frac{1}{N_{\text{classes}} - 1} g(\mathbf{S})\mathbf{I}. \quad (16)$$

Surprisingly, our empirical results obtained over a wide range of experimental conditions suggest that using the above approximate equalities gives very accurate results even in regimes where we include the off-diagonal entries of $\mathbf{WW}^\dagger$. In other words, the corrections obtained by including the off-diagonal entries are always marginal in our experiments.

## B  Training Aligned (TA) Networks

We now specialize the results in the previous section to the so-called TA networks [16][2]. TA networks are a class of analytically tractable models where one can explicitly calculate the quantities appearing in equations (8, 10, 13, and 16).

The key point is that TA networks are defined only by the choice of initialization of model parameters, and we are free to choose the initial values of these parameters to make the model solvable. Of course, in reality, deep learning practitioners do not have access to an oracle as in the student-teacher setup, so any initialization that assumes knowledge of the teachers' SVD is not feasible in practice. Nevertheless, simulations show that the intuition gained from TA models generalizes to networks initialized randomly.

For our TA model, we assume that we are using an SVD convention where the $\hat{\boldsymbol{U}}$ and $\hat{\boldsymbol{V}}$ are orthogonal matrices and the singular value matrix is rectangular with zeros off the main diagonal. Given the teacher's SVD $\beta\hat{\boldsymbol{\Sigma}} = \hat{\boldsymbol{U}}\hat{\boldsymbol{S}}\hat{\boldsymbol{V}}^\dagger$, we choose a set of orthogonal matrices $\{\boldsymbol{U}^{(l)}\}_{l=0}^L$ with $\boldsymbol{U}^{(l)} \in O(N_l)$, $\boldsymbol{U}^{(L)} := \hat{\boldsymbol{U}}$, $\boldsymbol{U}^{(0)} := \hat{\boldsymbol{V}}$, and set

$$\mathbf{W}_0^{(l)} = \boldsymbol{U}^{(l)}\boldsymbol{S}_0^{(l)}\boldsymbol{U}^{(l-1)\dagger}, \qquad \mathbf{W}_{<\,0}^{(l)} = \boldsymbol{U}^{(l-1)}\boldsymbol{S}_{<\,0}^{(l)}\hat{\boldsymbol{V}}^\dagger, \qquad \mathbf{W}_{>\,0}^{(l)} = \hat{\boldsymbol{U}}\boldsymbol{S}_{>\,0}^{(l)}\boldsymbol{U}^{(l)\dagger}$$

so that the student's initial composite weight matrix is

$$\mathbf{W}_0 = \mathbf{W}_{>\,0}^{(l)}\mathbf{W}_0^{(l)}\mathbf{W}_{<\,0}^{(l)} = \hat{\boldsymbol{U}}\boldsymbol{S}_{>\,0}^{(l)}\boldsymbol{S}_0^{(l)}\boldsymbol{S}_{<\,0}^{(l)}\hat{\boldsymbol{V}}^\dagger$$

Using the estimate in (16), the SGD update equations for the TA model at $t = 0$ read

$$
\begin{aligned}
\tau\frac{d}{dt}\boldsymbol{S}^{(l)}\bigg|_{t=0} &= \boldsymbol{S}_{>\,0}^{(l)\dagger}\hat{\boldsymbol{U}}^\dagger\Big[\mathbf{C}_{YX}(\beta\hat{\boldsymbol{\Sigma}}) - \mathbf{C}_{YX}(\mathbf{W}_0)\Big]\hat{\boldsymbol{V}}\boldsymbol{S}_{<\,0}^{(l)\dagger} \\
&\simeq \boldsymbol{S}_{>\,0}^{(l)\dagger}\Big[g(\hat{\boldsymbol{S}})\hat{\boldsymbol{S}} - g(\boldsymbol{S}_0)\boldsymbol{S}_0\Big]\boldsymbol{S}_{<\,0}^{(l)\dagger}
\end{aligned} \quad (17)
$$

The RHS of (17) is a diagonal matrix, so that

$$\boldsymbol{S}^{(l)}(\Delta t) \simeq \boldsymbol{S}_0^{(l)} + \frac{\Delta t}{\tau}\boldsymbol{S}_{>\,0}^{(l)\dagger}\Big[g(\hat{\boldsymbol{S}})\hat{\boldsymbol{S}} - g(\boldsymbol{S}_0)\boldsymbol{S}_0\Big]\boldsymbol{S}_{<\,0}^{(l)\dagger} + \mathcal{O}\left(\left(\frac{\Delta t}{\tau}\right)^2\right)$$

Thus, repeatedly iterating this construction gives, for arbitrary $t$,

$$\frac{d}{dt}\mathbf{W}^{(l)} = \boldsymbol{U}^{(l)}\frac{d}{dt}\boldsymbol{S}^{(l)}\boldsymbol{U}^{(l-1)\dagger}$$

with

$$\tau\frac{d}{dt}\boldsymbol{S}^{(l)} \quad\simeq\quad \boldsymbol{S}_>^{(l)\dagger}\Big[g(\hat{\boldsymbol{S}})\hat{\boldsymbol{S}} - g(\boldsymbol{S})\boldsymbol{S}\Big]\boldsymbol{S}_<^{(l)\dagger} \tag{18}$$

In other words, under the stated assumptions, SGD only modifies the singular values of the weights in each layer, leaving the singular vectors fixed at their initial values.

We henceforth drop the "$\simeq$" and write the equations as equalities. Specializing to the case where $L = 2$, equation (18) becomes

$$\tau\frac{d}{dt}\boldsymbol{S}^{(2)} \quad=\quad \Big[g(\hat{\boldsymbol{S}})\hat{\boldsymbol{S}} - g(\boldsymbol{S})\boldsymbol{S}\Big]\boldsymbol{S}^{(1)\dagger}, \qquad \tau\frac{d}{dt}\boldsymbol{S}^{(1)} = \boldsymbol{S}^{(2)\dagger}\Big[g(\hat{\boldsymbol{S}})\hat{\boldsymbol{S}} - g(\boldsymbol{S})\boldsymbol{S}\Big]$$

where $\boldsymbol{S} = \boldsymbol{S}^{(2)}\boldsymbol{S}^{(1)}$. If we define $\boldsymbol{s}^{(l)}$ as the vector consisting of the non-zero elements of $\boldsymbol{S}^{(l)}$, the previous equation reads

$$\tau\frac{d}{dt}\boldsymbol{s}^{(2)} \quad=\quad \Big[g(\hat{\boldsymbol{s}})\hat{\boldsymbol{s}} - g(\boldsymbol{s})\boldsymbol{s}\Big]\odot\boldsymbol{s}^{(1)}, \qquad \tau\frac{d}{dt}\boldsymbol{s}^{(1)} = \Big[g(\hat{\boldsymbol{s}})\hat{\boldsymbol{s}} - g(\boldsymbol{s})\boldsymbol{s}\Big]\odot\boldsymbol{s}^{(2)}$$

where, now $\boldsymbol{s} = \boldsymbol{s}^{(2)}\odot\boldsymbol{s}^{(1)}$. Consequently,

$$\tau\frac{d}{dt}\boldsymbol{s} = \tau\frac{d}{dt}\Big[\boldsymbol{s}^{(2)}\odot\boldsymbol{s}^{(1)}\Big] = \Big[g(\hat{\boldsymbol{s}})\hat{\boldsymbol{s}} - g(\boldsymbol{s})\boldsymbol{s}\Big]\odot\Big[\boldsymbol{s}^{(1)}\odot\boldsymbol{s}^{(1)} + \boldsymbol{s}^{(2)}\odot\boldsymbol{s}^{(2)}\Big]$$

Taking into account equation (9), we have

$$\boldsymbol{s}^{(1)}\odot\boldsymbol{s}^{(1)} = \boldsymbol{s}^{(2)}\odot\boldsymbol{s}^{(2)} + \text{constant}$$

where the constant term is determined by the choice of initial conditions. For simplicity, we pick the initial non-zero singular values to all have the same value so that the constant vanishes.

$$\boldsymbol{s}^{(1)}\odot\boldsymbol{s}^{(1)} = \boldsymbol{s}^{(2)}\odot\boldsymbol{s}^{(2)} \qquad\Rightarrow\qquad \boldsymbol{s} = \boldsymbol{s}^{(2)}\odot\boldsymbol{s}^{(1)} = \boldsymbol{s}^{(1)}\odot\boldsymbol{s}^{(1)}.$$

Thus, we finally obtain

$$\tau\frac{d}{dt}\boldsymbol{s} = \tau\frac{d}{dt}\Big[\boldsymbol{s}^{(2)}\odot\boldsymbol{s}^{(1)}\Big] = 2\Big[g(\hat{\boldsymbol{s}})\hat{\boldsymbol{s}} - g(\boldsymbol{s})\boldsymbol{s}\Big]\odot\boldsymbol{s}$$

which is equation (3) in our paper.

## C  Multitask Benefit

We will derive an expression for the multitask benefit $MT_{A\leftarrow B}$ in the most general setting, assuming Gaussian inputs (not necessarily iid) and linear activations, except for the softmax in the classifier. No other assumptions are required, and the result holds for models of any depth.

We will use the notation $\langle F(\vec{X})\rangle$ for the expectation of $F$ over the distribution of $\vec{X}$, where we assume that $\vec{X}$ is a centered Gaussian random vector with covariance $\mathbf{C}_X$.

The generalization error is obtained by computing

$$
\begin{aligned}
\mathcal{L} := \mathcal{L}_{\text{generalization}} \quad &= \quad -\left\langle \sum_{c=1}^{N_{\text{classes}}} \delta_{c,\overline{\mathbf{y}}(\vec{X})} \ln \mathbb{P}_c(\mathbf{W}\vec{X}) \right\rangle \qquad (19)\\
&= \quad -\sum_{c=1}^{N_{\text{classes}}} \left\langle \mathbb{P}_c(\beta \overline{\mathbf{W}}\vec{X}) \ln \mathbb{P}_c(\mathbf{W}\vec{X}) \right\rangle\\
&= \quad \langle \ln Z(\mathbf{W}\vec{X}) \rangle - \sum_{c=1}^{N_{\text{classes}}} \left\langle \mathbb{P}_c(\beta \overline{\mathbf{W}}\vec{X})[\mathbf{W}\vec{X}]_c \right\rangle
\end{aligned}
$$

where $Z(\vec{v}) := \sum_{c=1}^{N_{\text{classes}}} e^{v_c}$ for any vector $\vec{v} \in \mathbb{R}^{N_{\text{classes}}}$.

Note that we have replaced the noisy teacher's weights $\hat{\mathbf{\Sigma}}$ with the denoised teacher's weights $\overline{\mathbf{W}}$ since we test the model using the ground truth labels generated from the true distribution.

Using the same notation as in the main paper, we write $\mathbf{W}_A$ and $\widetilde{\mathbf{W}}_A$ for the parameters of task $A$ in the multitask setting and the single task baseline respectively. Thus, the generalization loss on the main task in the multitask setting is given by

$$
\mathcal{L}_{A|B} = \langle \ln Z(\mathbf{W}_A\vec{X}) \rangle - \lim_{\beta \to \infty} \sum_{c=1}^{N_{\text{classes}}} \left\langle \mathbb{P}_c(\beta \overline{\mathbf{W}}\vec{X})[\mathbf{W}_A\vec{X}]_c \right\rangle
$$

whereas the generalization loss for the baseline model trained on task A is

$$
\mathcal{L}_A = \langle \ln Z(\widetilde{\mathbf{W}}_A\vec{X}) \rangle - \lim_{\beta \to \infty} \sum_{c=1}^{N_{\text{classes}}} \left\langle \mathbb{P}_c(\beta \overline{\mathbf{W}}\vec{X})[\widetilde{\mathbf{W}}_A\vec{X}]_c \right\rangle
$$

The multitask benefit is obtained by computing

$$
MT_{A \leftarrow B} := \mathcal{L}_A - \mathcal{L}_{A|B} = \left\langle \ln \frac{Z(\widetilde{\mathbf{W}}_A\vec{X})}{Z(\mathbf{W}_A\vec{X})} \right\rangle + \lim_{\beta \to \infty} \sum_{c=1}^{N_{\text{classes}}} \left\langle \mathbb{P}_c(\beta \overline{\mathbf{W}}\vec{X})\left\{ [\mathbf{W}_A\vec{X}]_c - [\widetilde{\mathbf{W}}_A\vec{X}]_c \right\} \right\rangle
$$

Now, $\ln Z(\vec{v})$ is convex in $\vec{v}$ for any vector $\vec{v}$ (since the Hessian of $\ln Z$ is a positive definite symmetric matrix). Hence

$$
\begin{aligned}
\ln \left[ \frac{Z(\widetilde{\mathbf{W}}_A\vec{X})}{Z(\mathbf{W}_A\vec{X})} \right] \quad &\geq \quad (\widetilde{\mathbf{W}}_A - \mathbf{W}_A) \cdot \nabla_{\mathbf{W}_A} \ln Z(\mathbf{W}_A)\\
&= \quad \sum_{c=1}^{N_{\text{classes}}} \mathbb{P}_c(\mathbf{W}_A\vec{X})\left\{ [\widetilde{\mathbf{W}}_A\vec{X}]_c - [\mathbf{W}_A\vec{X}]_c \right\} \qquad (20)
\end{aligned}
$$

Interchanging $\mathbf{W}_A \leftrightarrow \widetilde{\mathbf{W}}_A$ yields

$$
\begin{aligned}
\ln \left[ \frac{Z(\widetilde{\mathbf{W}}_A\vec{X})}{Z(\mathbf{W}_A\vec{X})} \right] \quad &\leq \quad (\widetilde{\mathbf{W}}_A - \mathbf{W}_A) \cdot \nabla_{\widetilde{\mathbf{W}}_A} \ln Z(\widetilde{\mathbf{W}}_A)\\
&= \quad \sum_{c=1}^{N_{\text{classes}}} \mathbb{P}_c(\widetilde{\mathbf{W}}_A\vec{X})\left\{ [\widetilde{\mathbf{W}}_A\vec{X}]_c - [\mathbf{W}_A\vec{X}]_c \right\} \qquad (21)
\end{aligned}
$$

So, taking expectations in (20), we get

$$MT_{A \leftarrow B} \geq \lim_{\beta \to \infty} \sum_{c=1}^{N_{\text{classes}}} \left\langle \left[ \mathbb{P}_c(\mathbf{W}_A \vec{X}) - \mathbb{P}_c(\beta \overline{\mathbf{W}}_A \vec{X}) \right] \left\{ [\widetilde{\mathbf{W}}_A \vec{X}]_c - [\mathbf{W}_A \vec{X}]_c \right\} \right\rangle \qquad (22)$$

Similarly, taking expectations in (21) gives

$$MT_{A \leftarrow B} \leq \lim_{\beta \to \infty} \sum_{c=1}^{N_{\text{classes}}} \left\langle \left[ \mathbb{P}_c(\widetilde{\mathbf{W}}_A \vec{X}) - \mathbb{P}_c(\beta \overline{\mathbf{W}}_A \vec{X}) \right] \left\{ [\widetilde{\mathbf{W}}_A \vec{X}]_c - [\mathbf{W}_A \vec{X}]_c \right\} \right\rangle \qquad (23)$$

Using Gaussian integration by parts in (22, 23), we obtain, after some straightforward algebra that

$$MT_{A \leftarrow B} \geq \mathrm{Tr} \left( \left[ \mathbf{G}(\overline{\mathbf{W}}_A) \overline{\mathbf{W}}_A - \mathbf{G}(\mathbf{W}_A) \mathbf{W}_A \right] \mathbf{C}_X \left[ \mathbf{W}_A - \widetilde{\mathbf{W}}_A \right]^\dagger \right) \qquad (24)$$

and

$$MT_{A \leftarrow B} \leq \mathrm{Tr} \left( \left[ \mathbf{G}(\overline{\mathbf{W}}_A) \overline{\mathbf{W}}_A - \mathbf{G}(\widetilde{\mathbf{W}}_A) \widetilde{\mathbf{W}}_A \right] \mathbf{C}_X \left[ \mathbf{W}_A - \widetilde{\mathbf{W}}_A \right]^\dagger \right) \qquad (25)$$

where $\mathbf{G}$ is defined above in (13) via $\left\langle X_k \mathbb{P}_c(\mathbf{W} \vec{X}) \right\rangle = [\mathbf{G}(\mathbf{W}) \mathbf{W} \mathbf{C}_X]_{ck}$.

These expressions are completely general and do not assume TA initialization or make any other approximations other than the assumptions stated above, (*viz.* centered Gaussian random vectors with covariance $\mathbf{C}_X$, linear activations in the hidden layers and a softmax in the output layer).

It is also worth noting that the results hold for models of any depth since the $\mathbf{W}$'s refer to the composite weight of the entire network.

Specializing to the TA case, following Appendix B above, and elaborated further in Appendix C.1 gives the result quoted in our paper.

### C.1 Multitask Benefit for TA Networks

In order to address the multitask benefit for TA models, we need an extension of the single task analysis in Appendix B for multitask TA models. For simplicity, we consider the case where the data is drawn from a Gaussian distribution with $\mathbf{C}_X = \mathbf{I}$.

Recall that we defined the TA models by insisting that the student's initial weights have an SVD with the same singular vectors as in the teacher's SVD. The same definition applies here, so that if $\hat{\mathbf{\Sigma}}_{A/B} = \hat{U}_{A/B} \hat{S}_{A/B} \hat{V}_{AB}^\dagger$ denotes the teachers' SVDs for tasks A and B, then the SVDs for the students' initial weights for tasks A and B are respectively set to

$$\mathbf{W}_A(0) = \hat{U}_A \mathbf{S}_A \hat{V}_A^\dagger \qquad \text{and} \qquad \mathbf{W}_B(0) = \hat{U}_B \mathbf{S}_B \hat{V}_B^\dagger$$

Using the definition of task relatedness, $\boldsymbol{r}_{AB} = \hat{V}_B^\dagger \hat{V}_A$, in the previous expression gives

$$\mathbf{W}_A(0) = \hat{U}_A \mathbf{S}_A \hat{V}_A^\dagger \qquad \text{and} \qquad \mathbf{W}_B(0) = \hat{U}_B \mathbf{S}_B \boldsymbol{r}_{AB} \hat{V}_A^\dagger.$$

As in Appendix B, singular vectors corresponding to the composite weight matrices can be written as the Hadamard product of the singular vectors corresponding to the layerwise weights. For example, for a model with a single hidden layer, we have $\mathrm{diag}(\mathbf{S}_A) := \boldsymbol{s}_A^{32} \odot \boldsymbol{s}^{21}$ and $\mathrm{diag}(\mathbf{S}_B) := \boldsymbol{s}_B^{32} \odot \boldsymbol{s}^{21}$.

With these definitions, we can take the single task results for The TA model from Appendix B and extend them to two teachers to obtain

$$\tau \frac{d}{dt} s_A^{32} = s^{21} \odot \left( \hat{s}_A g(\hat{\Sigma}_A | \hat{U}_A) - s_A^{32} \odot s^{21} g(W_A | \hat{U}_A) \right)$$

$$\tau \frac{d}{dt} s_B^{32} = s^{21} \odot \left( \hat{s}_B g(\hat{\Sigma}_B | \hat{U}_B) - s_B^{32} \odot s^{21} g(W_B | \hat{U}_B) \right) r_{AB}$$

$$s^{21} \odot s^{21} = s_A^{32} \odot s_A^{32} + r_{AB} s_B^{32} \odot s_B^{32} \tag{26}$$

where $g(W | \hat{U}) := \hat{U}^\dagger G(W) \hat{U}$ for any pair of matrices $\hat{U}, W$.

Note that, as expected, if $r_{AB} = 0$, the dynamics for the second task is trivial and only the first task evolves non-trivially. Thus, to obtain the single task dynamics, we can simply look at the case $r_{AB} = 0$. This motivates the following definition.

1. Define $s(r_{AB})$ via the relation

$$s(r_{AB}) \odot s(r_{AB}) := s_A^{32} \odot s_A^{32} \odot \left[ s_A^{32} \odot s_A^{32} + r_{AB} s_B^{32} \odot s_B^{32} \right] \tag{27}$$

   This simply says that $\{s_\sigma(r_{AB})\}$ are the multitask student's singular values pertinent to executing task A.

2. If we set $r_{AB} = 0$ in (27), we recover the dynamics of the single task case. Therefore, if $\{\tilde{s}_\sigma\}$ denote the student's singular values when training on task A, we can identify

$$\tilde{s} = s(r_{AB} = 0)$$

To understand the difference between the multitask and single task case, we need to consider how $r_{AB}$ modifies the results in the single task case. The simplest way to do this is to study equation (26) perturbatively in $r_{AB}$ by plugging in the last line of (26) into the first line of (26). The mechanics of carrying out the perturbative expansion, while somewhat tedious, are straightforward and are left as an exercise for the motivated reader. The result of the exercise can be summarized as follows:

1. Let $\{\hat{s}_\sigma^A\}$ denote the singular values of the teacher corresponding to task A. By definition, the initial singular values corresponding to the multitask student's parameters for task A and those of the baseline single task student are identical. With these initial conditions, SGD dynamics yields, at all times,

$$
\begin{aligned}
s_\sigma(r_{AB}) &\geq s_\sigma(r_{AB} = 0) = \tilde{s}_\sigma & \text{whenever } \tilde{s}_\sigma|_{t=0} < \hat{s}_\sigma^A \\
s_\sigma(r_{AB}) &\leq s_\sigma(r_{AB} = 0) = \tilde{s}_\sigma & \text{whenever } \tilde{s}_\sigma|_{t=0} \geq \hat{s}_\sigma^A
\end{aligned}
\tag{28}
$$

   In other words, the effect of $r_{AB} > 0$ is to enhance either the growth rate or the decay rate of the student's singular values along the "principle components" of the noisy teacher.

2. Let $\{\bar{s}_\sigma^A\}$ denote the singular values of the noise-free teacher.

   (a) High SNR Case:
   When the SNR for task A is large, the singular vectors of the noise-free teacher are almost surely parallel to the singular values of the noisy teacher (see [25]). In this case,

$$\text{Tr}\left( \left[ G(\overline{W}_A) \overline{W}_A \right] \left[ W_A - \widetilde{W}_A \right]^\dagger \right) = \sum_{\sigma=1}^{\text{rank}_A} g(\bar{s}_A) \bar{s}_\sigma^A \left( s_\sigma(r_{AB}) - \tilde{s}_\sigma \right) \geq 0 \tag{29}$$

   where $\text{rank}_A := \text{rank}(\overline{W}_A)$ is the rank of the noise-free teacher. Consequently, equation (24) yields

$$
\begin{aligned}
MT_{A \leftarrow B} \geq &\sum_{\sigma=1}^{\text{rank}_A} \left[ g(\bar{s}_A) \bar{s}_\sigma^A - g(s(r_{AB})) s_\sigma(r_{AB}) \right] \left[ s_\sigma(r_{AB}) - \tilde{s}_\sigma \right] \\
&+ \sum_{\sigma > \text{rank}_A} g(s(r_{AB})) s_\sigma(r_{AB}) \left| s_\sigma(r_{AB}) - \tilde{s}_\sigma \right|
\end{aligned}
\tag{30}
$$

Thus, from the assertion in (15),

$$\|s(r_{AB})\| > \|\bar{s}_A\| \qquad \Rightarrow \qquad MT_{A \leftarrow B} > 0 \tag{31}$$

(b) Low SNR Case:

When the SNR for task A is small, the singular vectors of the noise-free teacher are almost surely orthogonal to the singular values of the noisy teacher (see [25]). In this case,

$$\text{Tr}\left( \left[ \mathbf{G}(\overline{\mathbf{W}}_A)\overline{\mathbf{W}}_A \right] \left[ \mathbf{W}_A - \widetilde{\mathbf{W}}_A \right]^{\dagger} \right) = 0 \tag{32}$$

Consequently, equations (24, 25) yield

$$\sum_{\sigma=1}^{N_{\text{classes}}} g(s(r_{AB}))s_{\sigma}(r_{AB})\left[ \tilde{s}_{\sigma} - s_{\sigma}(r_{AB}) \right] \leq MT_{A \leftarrow B} \leq \sum_{\sigma=1}^{N_{\text{classes}}} g(\widetilde{s})\tilde{s}_{\sigma}\left[ \tilde{s}_{\sigma} - s_{\sigma}(r_{AB}) \right]$$

$$\tag{33}$$

On the other hand, in the low SNR case, the singular values of the noisy teacher are essentially in the bulk of the MP sea (cf. [25]). Therefore, according to equation (28), the sign of the multitask benefit in the low SNR regime will depend on the size of the set $\{\sigma|\tilde{s}_{\sigma}|_{t=0} \geq \hat{s}_{\sigma}^A\}$ where the $\hat{s}_{\sigma}^A$ are drawn from the MP distribution. When this set is small, $MT_{A \leftarrow B}$ will tend to be negative. Conversely $MT_{A \leftarrow B}$ will tend to be positive if the size of the forementioned set is large.

3. Note that according equation (26), if $r_{AB} > 0$, then

   (a) increasing $\bar{s}_B$ (the SNR of task B) increases $\hat{s}_B$.
   (b) According to (15), increasing $\|\hat{s}_B\|$ decreases $g(\hat{s}_B)\hat{s}_B$, which in turn decreases the rate of growth of $s(r_{AB})$ relative to the rate of growth of $\widetilde{s}$ (cf. the second line in equation 26).

   Hence, an increase in $\hat{s}_B$ results in an overall increase of $\tilde{s}_{\sigma} - s_{\sigma}(r_{AB})$ and consequently an increase in $MT_{A \leftarrow B}$ in both the low SNR task A case following equation (33) and the high SNR task A case where the second term in (30) increases with $\left| \tilde{s}_{\sigma} - s_{\sigma}(r_{AB}) \right|$. In other words

   $$\text{increasing } \hat{s}_B\Big|_{r_{AB}>0} \Rightarrow \text{ an increase in } \left[ \tilde{s}_{\sigma} - s_{\sigma}(r_{AB}) \right] \Rightarrow \text{ an increase in } MT_{A \leftarrow B}$$

   $$\tag{34}$$

   Note that for small but nonzero values of $r_{AB}$ and very large values of $\|\hat{s}_B\|$, $g(\hat{s}_B)\hat{s}_B \to 0$ so that the second line of equation (26) leads to a rapid decay of $s_B^{32}$ towards zero, which in turn implies that $s(r_{AB}) - \widetilde{s}$ becomes negative following the third line of equation (26). Thus, $\left[ \tilde{s}_{\sigma} - s_{\sigma}(r_{AB}) \right] \to c_{\sigma} \leq 0$ for $\|\hat{s}_B\| \gg 1$. Consequently, regardless of the SNR for task A,

   $$MT_{A \leftarrow B} \to m \leq 0 \text{ for very large values of } \|\hat{s}_B\| \text{ and small } r_{AB} > 0. \tag{35}$$

4. We could also increase $r_{AB}$ with $\bar{s}_B > 0$ fixed.

   The third line of equation (26) shows that $s(r_{AB})$ monotonically increases with $r_{AB} > 0$. This in turn directly implies that the differences in the components of $\left[ s(r_{AB}) - \widetilde{s} \right]$ will all increase as $r_{AB}$ increases. Therefore, in the high SNR regime for task A, equation (30) immediately gives

   $$\text{an increase } r_{AB} \Rightarrow \text{ an increase in } MT_{A \leftarrow B} \tag{36}$$

5. Finally, we note that as $N_{\text{data}} \to \infty$, the empirical cross-covariance between the noisy labels and the input feature vectors converges to true cross-covariance between the noise-free labels and the input feature vectors as long as the noise level remains bounded by $N_{\text{classes}}/N_f$. In this case, the generalization loss and the training loss are almost surely equal so that

   $$\lim_{N_{\text{data}} \to \infty} \mathcal{L}_{A|B} \to \mathcal{L}_A \Rightarrow \lim_{N_{\text{data}} \to \infty} MT_{A \leftarrow B} \to 0. \tag{37}$$

# D  Task Relatedness and Multitask Results

## D.1  Multitask full results

Figure A1: Multitask benefit in student networks trained on data that varied along 4 independent variables: 1) number of main task training points (rows), 2) main task signal-to-noise ratio (SNR) (columns), 3) auxiliary task SNR (x-axes), and 4) auxiliary task relatedness (individual lines). Each line shows the mean benefit over 5 random seeds and the shaded region shows the standard error of the mean. Multitask benefits > 0 indicate that student network performs better when trained with additional auxiliary task data. MT benefit is correlated with task relatedness and SNR for related tasks, yet negatively correlated with SNR for unrelated tasks. This data is summarized in Figure 2.

Figure A2: Multitask benefit when trained on rank 3 teachers. The data is arranged as in Figure A1 and shows very similar trends as in the rank 10 case.

Figure A3: Multitask benefit when trained on increasing levels of auxiliary task data. The data is arranged as in Figure A1 and shows greatly improved performance with larger amounts of auxiliary data. This data is summarized in Figure 3, left panel.

Figure A4: Multitask benefit when training deeper networks with ReLU nonlinearities. The data is arranged as in Figure A1 and shows qualitatively similar results to linear networks. This data is summarized in Figure 3, right panel.

## Footnotes

[2]Our definition of TA networks differs slightly from the TA networks in [16].