[Reviews · NeurIPS 2019]

Reviewer 1



Overall the paper is clear and easy to follow. The proofs and main derivations are all in the appendix and while I did not thoroughly check them they are relatively straightforward. The idea of analyzing the training dynamics by taking a tractable simple model, making approximations, and deriving both qualitative and quantitative conclusions which can be validated experimentally is quite interesting. The particular approach adopted here to analyze multitask learning in particular is novel.

Reviewer 2



Originality: The theoretical results on single and multitask learning in classification networks are novel as in previous literature the focus was mainly on regression networks. The key analysis are although inspired by [16] but tackling the cross entropy loss function in this setting is original. Quality: The theoretical derivation of dynamics of generalization of both single task and multitask classification networks and its comparison to empirical results thorugh well thought experiments makes this paper a good contribution. The results of deeper non linear networks showing a similar trends to shallow linear networks makes this contribution much stronger. Clarity: The paper is well-written, the theory clearly described and presentation of results through descriptive figures. Significance: The contribution is important for understanding generalization dynamics of classification networks. The theoretical results and their empirical validation on generalization dynamics in multitask setting is very relevant in complex setups where multiple auxiliary tasks can aid in learning a new tasks.

Reviewer 3



## Overall The paper presents an interesting analysis of linear classification networks using the student-teacher framework. The experiments on multitask learning are informative. I wish the experiments and theory were a bit more integrated. See my comments below for more details. ## Writing + The paper is clearly written. The authors moved a lot of details to the appendix while keeping the main conclusions in the main submission to ease understanding. - Often, the authors state results without referring to the corresponding equation number in Appendix. Here are some examples: (a) L181-184 what equation shows (s_A - \tilde{s_A}) depends on the said 4 things; (b) L185-186 when labelled data is scarce why is (\bar{s_A*g(s_A)}-\tilde{s_A*g(s_A)} < 0; (c) L189-190 why does (\bar{s_A*g(s_A)}-\tilde{s_A*g(s_A)} tend to 0 when training data is abundant. It's not obvious where these results are coming from or why they are true. - The key takeaways or results need to be more explicitly stated - The title reads "a statistical physics approach" but its unclear what techniques are borrowed from physics. The authors only refer to a connection to the physics of the disordered systems loosely. If this connection is important, please state the result in physics describing the physical system and variables involved, and then draw analogies to learning dynamics in the network. ## Empirical Evaluation + The experiments on multitask learning are illustrative + Fig 2 quite cleverly manages to show the relation between 5 different variables - Having said that, the figure is quite complicated and the main conclusions are getting a bit lost in a wall of plots. My suggestion would be to break Fig 2. into smaller plots considering only a subset of the variables necessary to make the point (e.g. by fixing the SNRs to a certain value when trying to see the effect of training data size). - The multitask experiments seem a bit disconnected from the analysis done earlier. The authors are encouraged to refer to the relevant equations in the experiments and point out how the empirical conclusions match the theory. ============================================================ After reading the other reviews and rebuttal I think the paper should be accepted. My main concerns were related to writing and based on the rebuttal I trust the authors to update the manuscript accordingly. This is the first theoretically motivated analysis of multitask learning that I have come across and I would love to see more work building up on this. For any revised manuscript, my recommendation would be to include the following: 1. Tab 1 from the rebuttal (also reference the relevant equation in “analytical explanation” column) 2. improved explanation of Derrida’s Random Energy Model to make the paper self contained (most readers would be unfamiliar with it; ok to put this in Appendix). 3. add references to relevant equations in the appendix and consider adding the key equations to the main manuscript.

[Author Response · NeurIPS 2019]

# Generalization in multitask deep neural classifiers: a statistical physics approach

We would first like to thank all three reviewers for their thorough, constructive and considered reviews. Each of them showed a good understanding and appreciation of our work, and we are glad they found it to be a clear and interesting approach to an important problem in the field. Even after allowing for many simplifications to make our model theoretically tractable, there remain several technically involved derivations that were difficult to describe clearly in our attempts at linking the analytical and experimental results. The reviewers have correctly pointed out instances where we could do a better job describing our findings. We will go through these below and will incorporate these suggestions within the final manuscript.

**Ties to statistical physics**: The title was intended to convey the fact that the techniques employed in our analysis are extensions of those used by physicists to analyze disordered systems. As alluded to in the opening to section 2 and Appendix A, our model is a nonequilibrium variant of Derrida's Random Energy Model. The stochasticity in the data is the analog of "quenched disorder" in a spin-glass while the loss plays the role of the free energy. We are, of course, not the first to use such an analogy (e.g. references 4-7 in the appendices to our paper). This correspondence provided invaluable intuition that rendered calculating expectations in our model tractable, thereby allowing us to draw strong conclusions on the training dynamics. We will update the final manuscript to describe this analogy more explicitly.

**On increasing the sophistication of the teacher**: We're assuming that by "a more complicated ground truth", the reviewer means either using a deep, non-linear neural net as a teacher or directly using the empirical distribution of the data. In the former case, our techniques carry over to deep teachers with ReLU activations at the expense of additional technical baggage and the ad hoc assumption that ReLU transformations preserve the covariance of a distribution up to an overall scaling factor. As for the latter case, the primary challenge lies in properly defining task relatedness for a given dataset. We have thought quite a bit about how to run these experiments using MNIST, but have been blocked on defining a robust measure of task-relatedness. This is a non-trivial problem that, to the best of our knowledge, has not been addressed in the literature. As such, this is still a matter of active research. One key element of the definition in the manuscript is that it can be varied independent of noise level. Retaining relatedness as a factor independent of noise unfortunately excludes many potential definitions (e.g. angle between tasks in an embedding space). Thus, we feel that such an extension presents many original challenges and should remain outside the scope of the current study.

**Rank of the teachers/students**: Our results hold for arbitrary rank as long as $\text{rank}_{\text{student}} \geq \text{rank}_{\text{teacher}}$. We highlighted the rank-1 case solely to ease visualization of the dynamics. We should have clarified this in the paper and will do so in the final version.

**Improving the exposition of the derivations**

*Conditions claimed in L181-184*: We will amend the manuscript to indicate that the equation directly preceding eqn. A:20 in the appendix implies the conditions stated. We will re-label the equations accordingly.

*L185-186 & L189-190*: why is $\overline{s}_A g(\overline{s}_A) < \widetilde{s}_A g(\widetilde{s}_A)$ when labeled data is scarce and why does $\overline{s}_A g(\overline{s}_A) \to \widetilde{s}_A g(\widetilde{s}_A)$ when training data is abundant?

Briefly, we approximate the matrix $G(\mathbf{W}) \simeq g(\mathbf{W})\mathbb{I}$, where $g(\mathbf{W})$ is a scalar function (see Appendix B). This is strictly true for $N_{\text{data}} \to \infty$. The leading order correction is of $\mathcal{O}(\frac{1}{N_{\text{data}}})$. For finite size datasets, assuming that $G(\mathbf{W})$ evolves on a much slower time scale than $\mathbf{W}$, integrating eqn. (11) in Appendix B yields $\dfrac{\widetilde{s}_A g(\widetilde{s}_A)}{\overline{s}_A g(\overline{s}_A)} = \dfrac{1}{1 - \frac{|\text{const}|}{N_{\text{data}}} + \mathcal{O}\left(\frac{1}{N_{\text{data}}^2}\right)}$

as the fixed point of the dynamics. This gives the quoted results. We will include a detailed argument in the appendix. More generally, we will update the main paper to better reference the derivations in the appendix.

**Clear takeaways and link to experiments**

The remaining suggestions focused on clarifying the key takeaways and linking the experimental and analytical findings. We now summarize our analytical results in a table (a subset of rows are shown in table 1 due to space constraints) and will clearly reference it when discussing the corresponding results in the experimental section. Finally, Figure 2 was likely too ambitious, so we have redone it to highlight subsets of variables as we did in Figure 3. The original Figure 2 will be moved to the appendix for completeness.

Table 1: A sampling of the key takeaways

| independent variables | | | | |
|---|---|---|---|---|
| $r_{AB}$ | $\overline{s}_B$ | $N_{\text{data}}$ | effect on $MT_{A \leftarrow B}$ | analytical explanation |
| 0 | any | any | 0 | $s_A = \widetilde{s}_A$ |
| $> 0$ | $\nearrow$ | any | $\nearrow$ | $(s_A - \widetilde{s}_A) \searrow$ as $\overline{s}_B \nearrow$ |
| $r_{AB} \nearrow (0 < r_{AB} \ll 1)$ | any | limited | $\nearrow$ | $(s_A - \widetilde{s}_A) \searrow$ as $r_{AB} \nearrow$ |
| any | any | abundant | small | $\widetilde{s}_A g(\widetilde{s}_A) \to \overline{s}_A g(\overline{s}_A)$ |

[Meta-Review · NeurIPS 2019]

This paper is a nice combination of theoretical understanding and simple experiments to verify it in the case of multitask learning in neural nets. Given that there is not much known in this space, this work can be impactful. I suggest authors to add a few multi-task experiments with real datasets to verify their understanding.